# Testing "efficient supply chain propositions" using topological characterization of the global supply chain network

**Abhijit Chakraborty**[1¤a¤b], **Yuichi Ikeda**[2]*

**1** Graduate School of Simulation Studies, The University of Hyogo, Kobe, Japan, **2** Graduate School of Advanced Integrated Studies in Human Survivability, Kyoto University, Kyoto, Japan

¤a Current address: Complexity Science Hub Vienna, Vienna, Austria
¤b Current address: Advanced Systems Analysis, International Institute for Applied Systems Analysis, Laxenburg, Austria
* ikeda.yuichi.2w@kyoto-u.ac.jp

**Data Availability Statement:** The Standard & Poor's "Capital IQ platform" data was used in the current study. Standard & Poor's offers various company-level data in accordance with customers' demands, such as detailed company descriptions,

## Abstract

In this paper, we study the topological properties of the global supply chain network in terms of its degree distribution, clustering coefficient, degree-degree correlation, bow-tie structure, and community structure to test the efficient supply chain propositions proposed by E. J.S. Hearnshaw *et al*. The global supply chain data in the year 2017 are constructed by collecting various company data from the web site of Standard & Poor's Capital IQ platform. The in- and out-degree distributions are characterized by a power law of the form of $\gamma_{in}$ = 2.42 and $\gamma_{out}$ = 2.11. The clustering coefficient decays $\langle C(k) \rangle \sim k^{-\beta_k}$ with an exponent $\beta_k$ = 0.46. The nodal degree-degree correlations $\langle k_{nn}(k) \rangle$ indicates the absence of assortativity. The bow-tie structure of giant weakly connected component (GWCC) reveals that the OUT component is the largest and consists 41.1% of all firms. The giant strong connected component (GSCC) is comprised of 16.4% of all firms. We observe that upstream or downstream firms are located a few steps away from the GSCC. Furthermore, we uncover the community structures of the network and characterize them according to their location and industry classification. We observe that the largest community consists of the consumer discretionary sector based mainly in the United States (US). These firms belong to the OUT component in the bow-tie structure of the global supply chain network. Finally, we confirm the validity of Hearnshaw *et al*.'s efficient supply chain propositions, namely Proposition S1 (short path length), Proposition S2 (power-law degree distribution), Proposition S3 (high clustering coefficient), Proposition S4 ("fit-gets-richer" growth mechanism), Proposition S5 (truncation of power-law degree distribution), and Proposition S7 (community structure with overlapping boundaries) regarding the global supply chain network. While the original propositions S1 just mentioned a short path length, we found the short path from the GSCC to IN and OUT by analyzing the bow-tie structure. Therefore, the short path length in the bow-tie structure is a conceptual addition to the original propositions of Hearnshaw.

corporate structures, global supply chain data, global ownership data, etc in the "Capital IQ platform". But restrictions apply to the availability of these data, which were used under license for the current study, and so are not in public domain. The data is available from Standard & Poor's website: \url{https://www.spglobal.com/marketintelligence/en/solutions/sp-capital-iq-platform}, after signing a contract with Standard & Poor's and paying a data usage fee to Standard & Poor's.

**Funding:** The present study was supported by the Ministry of Education, Science, Sports, and Culture, Grants-in-Aid for Scientific Research (B), Grant no. 17KT0034 (2017-2019) to YI and Exploratory Challenges on Post-K computer (Studies of Multi-level Spatiotemporal Simulation of Socioeconomic Phenomena) to AC.

**Competing interests:** The authors have declared that no competing interests exist.

# Introduction

National economies are linked through international trade so the globalized economy forms a large economic complex network characterized by strong links, i.e., interactions due to increasing level of trade between countries worldwide. If we view the globalized world economy with a high resolution or a microscopic view, we might notice that this large economic network consist of a global supply chain comprised of a large number of firms. A variety of collective motions exist in natural and social phenomena. Some examples of the collective motions of the global economy include the synchronization of the business cycle, global economic crises, and chain bankruptcies. These collective motions occur due to the strong interactions between constituent elements. Thus, it is expected that various collective motions will emerge in the globalized world economy under the condition of trade liberalization, i.e. globalization.

Several review papers have been published that examine study of supply chain network, M. A. Bellamy *et al*. [1] categorized the study of supply chain networks into three themes: network structure (i.e., system architecture), network dynamics (i.e., system behavior), and network strategy (system policy and controls). They listed important factors that characterize supply chain networks. For example, factors that characterize network structures include node-level, network-level, and link-level properties. The factors characterizing network dynamics are stimuli, phenomena, and sustainability. The factors characterizing network strategy are scope, intent, and governance. S. Perera *et al*. [2] surveyed the methodologies for the purpose of modeling topology and robustness. They pointed out the limitation of the preferential attachment growth model for explaining the characteristics of the supply chain networks and emphasized the importance of fitness-based growth models [3] for explaining the observed topological characteristics. Notable phenomena regarding supply chain networks include not only resilience against random failure and targeted attacks but also collective motion such as cascading failure or chain bankruptcy. Y. Fujiwara studied chain bankruptcy by analyzing the supply chain and bankruptcy data and Y. Ikeda developed an agent-based model and constructed realistic simulations of the chain bankruptcies caused by the failure of a single firm [4]. K. J. Mizgier *et al*. [5] studied the dynamics of default processes in supply chain networks using an agent-based model. Based on a simulation, they discussed the implications of such dynamics on risk management and policy making. L. Tang *et al*. [6] developed a theoretical cascading failure model considering the interdependence of firms in the supply chain network. They observed a sudden collapse in the interdependence of supply chain networks. T. Mizuno *et al*. [7] have analyzed a large set of global supply chain data. They investigated three different types of networks: a customer-supplier network, a licensee-licensor network, and a strategic alliance network. The degree distributions of all three networks showed scale-free properties characterized by a power law tail. They also observed that all three networks showed average path lengths of around six. They further studied the community structures of undirected versions of networks using modularity maximization techniques [8].

In addition to these studies, E. J.S. Hearnshaw *et al*. [9] studied the supply chain networks using a complex network approach and proposed the following nine propositions:

- S1: *Efficient supply chain systems show a short characteristic path length*

- S2: *The nodal degree distribution of efficient supply chain systems follows a power law as indicated by the presence of hub firms*

- S3: *Efficient supply chain systems demonstrate a high clustering coefficient*

- S4: *The growth of efficient supply chain systems follows the "fit-gets-richer" mechanism*

- S5: *The power law degree distribution of efficient supply chain systems is truncated*

- S6: *The link weight distribution of efficient supply chain systems follows a power law*

- S7: *Efficient supply chain systems demonstrate a pronounced community structure with overlapping boundaries*

- S8: *The fitness of hub firms determines the resilience of supply chain systems against both random disturbances and targeted attacks*

- S9: *Resilient supply chain systems demonstrate a power law distribution for link-weights*

In E. J.S. Hearnshaw *et al*.'s study, the term "efficient supply chain" was used to characterize the fact that a variety of resources such as financial, human, technological, and physical resources were effectively used in supply chains. These nine propositions S1 through S9 are related to path length, power law degree distribution, clustering coefficients, preferential attachment growth mechanism, truncated power law connectivity distribution, power law distribution of node strength, community structure with overlapping boundaries, resilience against random failure and targeted attack, and core-periphery structure, respectively. They attempt to explain the various functions of a supply chain according to the structural characteristics of supply chain networks. It is, however, noted that the nine propositions were proposed by a theoretical study based on a survey of the existing literature using only small-scale empirical data. Its empirical verification is rather weak and therefore a large-scale empirical study using more comprehensive data would be highly desired and contribute significantly to the extant literature on supply chain networks.

To understand the globalized world economy and make effective policy recommendations, studying the global supply chain, international trade, business cycles, and economic growth by analyzing global data using network scientific methodology is indispensable. In this paper, we focus on the topological properties of the global supply chain network to empirically test the efficient supply chain propositions put forth by E. J.S. Hearnshaw *et al*. We contribute to the literature a verification of these propositions by conducting a large-scale empirical analysis of global supply chain data. This study on the topological properties of the global supply chain network is a first step in understanding the globalized world economy under a microscopic view. We study degree distribution, clustering coefficient, and degree-degree correlation in the global supply chain network. A bow-tie structure was uncovered based on the supply of goods and services along directed links. Then, we uncovered the community structures of the network using the map equation method and characterized them according to their locations and industry classifications. Furthermore, the composition of the communities in terms of bow-tie components is analyzed. Finally, we investigate the validity of E. J.S. Hearnshaw *et al*.'s nine propositions regarding the efficiency of supply chain networks [9] based on the results obtained regarding the topological properties of the global supply chain network.

Our paper is organized as follows. In the "Data" section we briefly describe the global supply chain network data used in this study. The data in the year 2017 were collected from Standard & Poor's Capital IQ platform. In the "Methods" section, methodologies for the identification of the bow-tie structure, community detection, and the over-expression of bow-tie components are explained. In the "Results" section, the results obtained from the analysis of global supply chain network data—basic structural properties, bow-tie structures, community structures, and over-expression of bow-tie components—are explained using figures and tables. Finally, we investigate the validity of the nine propositions based on the obtained results on the topological properties of global supply chain networks. To close, this paper concludes in the "Conclusions" section.

**Table 1. Types of business relationships for all firms.**

| Relationship | #Links | Ratio (%) |
|---|---|---|
| Supplier | 849,223 | 59.0 |
| Creditor | 465,412 | 32.3 |
| Landload | 76,908 | 5.3 |
| Licensor | 47,558 | 3.3 |
| Total links | 1,439,101 | 100.0 |

## Data

The global supply chain data in the year 2017 were constructed by collecting various firm data from the web site of Standard & Poor's (S&P) Capital IQ platform. The data include firm identification (ID), name, country, and location, primary industry, and sector as node information. Industrial classification is based on the Global Industry Classification Standard (GICS), which was developed by Morgan Stanley Capital International and the S&P. Firms from 11 sectors and 158 primary industries are located in 206 countries as listed in S1–S3 Tables in S1 Appendix.

The data also include types of business relationships between suppliers and customers as link information. Although there are a variety of business relationships types including suppliers, creditor, franchisor, licensor, landlord, lessor, auditor, transfer agent, investor relations firm, and vendor, most of the relationships are of the supplier and creditor types. Here, the supplier type indicates a firm that provides the products or services and the creditor type indicates a private, public, or institutional entity that loans funds to others entities.

In Table 1, the types of business relationships of all firms are summarized. We note that the links in the dataset are dominated by supply chain business relationships. In Table 2, the supplier types of all firms are summarized. We note that suppliers are dominated by private and public firms. Therefore, the characteristics of the dataset reflect the nature of the global supply chain network.

We preprocessed the raw data in the following way. We removed all creditor relationships, isolated nodes, parallel links, and self-loops from the dataset being used in our analysis. The total number of firms and directed links used in our analysis are 437, 453 and 948, 247, respectively. The number of firms and total revenue of firms in each country are listed in S1 Table in S1 Appendix. Firm distributions for the different sectors are listed in S2 Table in S1 Appendix. It is noted that S2 Table in S1 Appendix represents the data after preprocessing, while the Table 2 represents the raw data. Hereafter, the preprocessed data is referred to as the global supply chain data in the year 2017.

Aggregated revenue is compared to the gross domestic product (GDP) of each country, as shown in Fig 1. These statistics provide evidence for the goodness of data coverage of our

**Table 2. Types of suppliers for all firms.**

| Firm Type | #firms | Ratio (%) |
|---|---|---|
| Private firm | 830,915 | 58.6 |
| Private fund | 32,180 | 2.3 |
| Private investment firm | 20,164 | 1.4 |
| Public firm | 533,910 | 37.7 |
| Total firms | 1,417,169 | 100.0 |

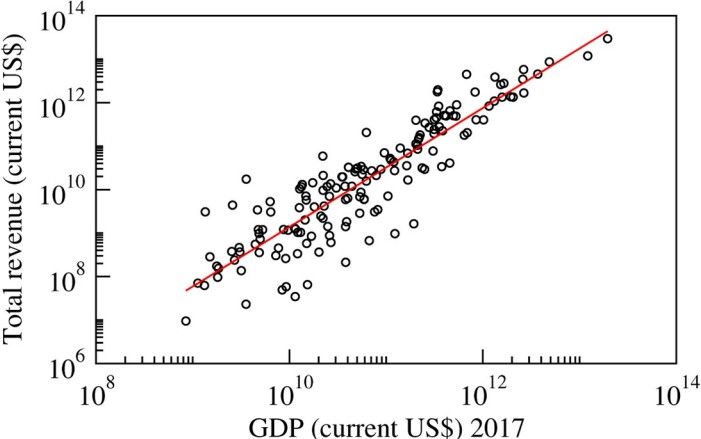

**Fig 1. Total revenue generated by the firms in a country plotted against the country's GDP in current US $ for the year 2017.** The vertical axis is aggregated for firms of each country in the global supply chain data. The red line represents the best power law fit to the data of the form revenue $= 2.81 \times 10^{-5} \text{GDP}^{1.369}$.

global supply chain data. GDP data were collected from https://data.worldbank.org/ and are therefore in the public domain.

## Methods of topological characterization

In order to test Hearnshaw *et al.*'s efficient supply chain propositions, the topological properties of the global supply chain network are empirically studied from three different angles: their node-level characteristics, mesoscopic structural characteristics, and flow characteristics. The analysis from these three different angles is required to test the efficient supply chain propositions and obtain a coarse-grained description of the global supply chain network in terms of bow-tie and community structures.

### Node-level characteristics

As the supply chain network is directed by nature, one can define in and out degrees for the nodes. The nodal in-degree is defined as the number of incoming links to a node and the out-degree is the total number of outgoing links from that node. The most basic node-level characteristic is the degree distribution for the nodal in-degrees $k_{in}$ and nodal out-degrees $k_{out}$. The second node-level characteristic is the clustering coefficient. The clustering coefficient, a measure of three-point correlation, reflects the cliquishness among the neighbors of a node. The clustering coefficient $C_i$ of $i$-th node is defined as [10],

$$C_i = 2E_i/k_i(k_i - 1), \tag{1}$$

where $E_i$ is the number of links between all the $k_i$ neighbors of $i$. We have measured the clustering coefficient $<C(k)>$ averaged over the subset of nodes of degree $k$. Further one can measure more complex clustering coefficient considering the directed nature of links [11, 12]. However, we only calculate the simple undirected clustering coefficient to show the basic structural property of the network. The third node-level characteristic is the degree-degree correlation. The average nearest neighbor degree $\langle k_{nn}(k) \rangle$ increases as a function of degree $k$ for an assorative network and decreases for a disassortative network. We have used an undirected version of the network for measuring the clustering coefficient and average nearest neighbor degree to

show the presence of hierarchical structure and degree mixing property of the network as the basic structural properties.

## Mesoscopic structural characteristics

The bow-tie structure [13] is uncovered from the giant weakly connected component (GWCC) based on the supply of goods and services (i.e., money moves in the opposite direction) along the directed links. The definitions of the different regions of the bow-tie structure are as follows:

- The giant strongly connected component (GSCC): The largest region in which any two nodes are reachable through the directed path.

- IN components: The nodes from which the GSCC are reachable through directed paths.

- OUT components: The nodes that are reachable from the GSCC through directed paths.

- Tendrils (TE): The rest of the nodes in the GWCC.

We apply a breadth-first search algorithm to detect the different components of the bow-tie structure.

## Flow characteristics

Empirical networks are generally non-homogeneous with a high local link density. Community detection captures highly connected groups of nodes as modules. Modularity maximization [14] is one of the popular methods for detecting communities. With this method, one maximizes the modularity index. Modularity is defined as the fraction of intra-community links, subtracting the expected fraction, given a random distribution. However, this method suffers from a resolution limit problem [15] when applied to large networks. Despite this problem, the modularity maximization is still an effective method for detecting communities in moderate sized undirected and directed networks. We use the modularity maximization to detect communities of over-expression networks in our study.

The map equation method [16] detects communities using the flow dynamics of random walkers on the network. We use the map equation method for our analysis of the global supply chain network since it is a directed network of suppliers and customers in which link represents the flow of goods. This method is one of the most suitable community detection techniques for detecting communities in a network [17] based on the flow of random walkers that does not suffer from the resolution limit because our main interest in community detection is in extracting flow characteristics. This minimizes the per step the average description length $L(C)$ of a random walker on the network as defined below:

$$L(C) = q_\curvearrowright H(C) + \sum_{i=1}^{m} p_\circlearrowleft^i H(\mathcal{P}^i) \ . \tag{2}$$

where $q_\curvearrowright$ and $H(C)$ represent the probability and Shannon entropy, respectively for the inter-community movement of the random walker. $p_\circlearrowleft^i$ is the probability that a random walker leaves node $i$ and $H(\mathcal{P}^i)$ is the entropy of intra-community movement.

## Coarse-grained description

Communities are ubiquitous in empirical networks. Communities are formed according to similarities in node attributes. For examples locations and sectors are key attributes for the formation of communities in the Japanese supply chain network [18, 19], in protein-protein

interaction networks, biological functions form the basis of the community structure [20]. The node attribute within communities is the key to obtaining a coarse-grained description of the community structure of the global supply chain network.

The composition of communities in terms of their bow-tie components is analyzed using the over-expression of node attributes. To measure the over-expression of attributes in a community, we follow the method used by Tumminello *et al.* [21]. In this method, the probability that $X$ randomly selected nodes in a community $C$ of size $N_C$ has the attribute $A$ is calculated using the following hyper-geometric distribution

$$H(X|N, N_C, N_A) = \frac{\binom{N_C}{X}\binom{N - N_C}{N_A - X}}{\binom{N}{N_A}},$$

where $N_A$ is the total number of nodes in the network with attribute $A$. The $p$-value $p(N_{C,A})$ for the $N_{C,A}$ nodes with attribute $A$ in the community $C$ can be obtained from the following expression:

$$p(N_{C,A}) = 1 - \sum_{X=0}^{N_{C,A}-1} H(X|N, N_C, N_A).$$

Attribute $A$ is over-expressed when $p(N_{C,A})$ is lower than the threshold value $p_c$. Since this is a multiple hypotheses test, one must choose $p_c$ appropriately to exclude generating false positives. We set $p_c = 0.01/N_{A'}$ in line with [21], which accomodate the Bonferroni correction [22]. Here, $N_{A'}$ indicates the total number of distinct attributes of all nodes of the network.

## Results

### Node-level characteristics

We observe that the probability density distributions for both nodal in and out degrees feature a heavy-tail in which the tail of the distributions is characterized by a power law of the form $P(k_{in/out}) \sim k^{-\gamma_{in/out}}$ with $\gamma_{in} = 2.42$ and $\gamma_{out} = 2.11$, respectively, as shown in Fig 2(a) and 2(b). It is, however, noted that we are not claiming that the entire distribution follows a power law distribution. We measured the exponent from the slope of the straight portion of the intermediate range of degrees. The tail region of both distributions seem to be truncated due to the finite system size. The power law degree distribution has also been observed in past investigations of empirical supply chain network data [7, 23–25]. Degree distribution plays a pivotal role in shock propagation between nodes. High asymmetry in degree distribution can result in system wide aggregate fluctuations due to idiosyncratic shocks to large firms [26]. It has been argued in the literature that such heavy-tail distributions of nodal degrees arises due to the so-called "rich-get-richer" mechanism [27, 28]. Like with the rich-get-richer principle, here, large firms have more customers and suppliers than do small firms.

In most real-world networks, the average clustering coefficient is a decaying function of degrees having the form $\langle C(k)\rangle \sim k^{-\beta_k}$ with $\beta_k \leq 1.0$. We observe that the clustering coefficient of the supply chain network decays with an exponent $\beta_k = 0.46$ as shown in Fig 2(c), which indicates the presence of a hierarchical structure.

The average degree of the neighbors of node $i$ that capture the nodal degree-degree correlation is defined as $k_{nn,i} = \sum_j k_j/k_i$, where the $j$ runs over all $k_i$ neighbors of $i$. For nodes with degree $k$, $\langle k_{nn}(k)\rangle = \sum_{k_i=k} k_{nn,i}/N_k = \sum_{k_1} k_1 P(k_1|k)$, where $N_k$ is the number of nodes having degree $k$. $\langle k_{nn}(k)\rangle$ increases with $k$ for an assortative network and decreases for a disassortative

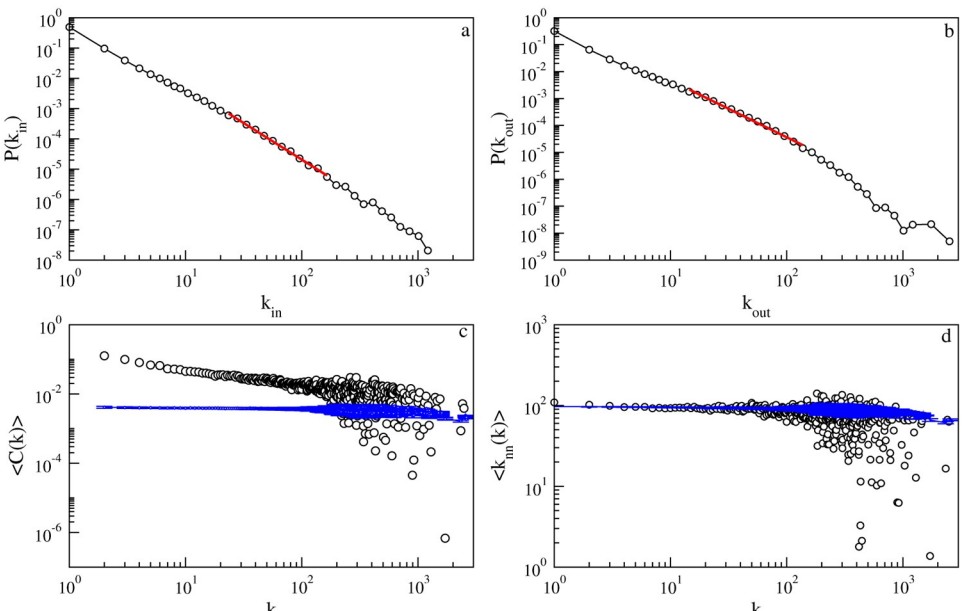

**Fig 2. Structural properties of the global supply chain network.** Probability density distributions $P$ of: (a) nodal in-degrees $k_{in}$ and (b) nodal out-degrees $k_{out}$. Variations of (c) the clustering coefficient $C(k)$ as a function of degree $k$ and (d) the average nearest-neighbor degree $\langle k_{nn}(k) \rangle$ as a function of degree $k$. Logarithmic binning of the horizontal axis is used in (a) and (b). Red lines represent the best power-law fit to the data. Blue lines in (c) and (d) represent the results for degree-preserved random networks in which the average is taken over 100 such uncorrelated networks. The standard deviations are shown as blue vertical lines in (c) and (d).

network. In the absence of nodal degree-degree correlation $\langle k_{nn}(k) \rangle$ remains constant. As can be seen in Fig 2(d), $\langle k_{nn}(k) \rangle$ does not depend on $k$ and remains more or less constant with $k$, indicating the absence of nodal degree-degree correlation.

Further, the statistical significance of these results is tested by comparing them with the results of randomized degree-preserving networks [29]. The standard deviations are shown as blue vertical lines in Fig 2(c) and 2(d). The difference between the results obtained from the data and results from the randomized degree-preserving network is evident and, therefore, the obtained results are statistically significant. The clustering coefficients of randomized networks show $C(k) \sim$ constant, as expected. The variation of $\langle k_{nn}(k) \rangle$ with $k$ matches nicely with the case of degree-preserving randomized networks, which further supports the absence of nodal degree-degree correlations in the empirical network.

We examine the connected components when the network is viewed as an undirected network. The largest connected component of the network is known as the GWCC. As can be seen in Fig 3, the network consists of a very large GWCC with $N = 407, 527$ nodes, and $L = 927, 316$ links. Using a breadth-first search, we calculate the average path length in the GWCC by calculating the shortest paths between all pairs of nodes. The average path length is found to be 5.370, reflecting the small-world nature of the global supply chain network. While the GWCC contains 93.16% of nodes of the network, the rest of the components are very small. In the following subsequent sections, we investigate only the GWCC of the network.

## Mesoscopic structural characteristics: Bow-tie structure

We detect the bow-tie components in the GWCC of the global supply chain network. The number of firms in each component is shown in Table 3. Here, "Ratio" refers to the ratio of

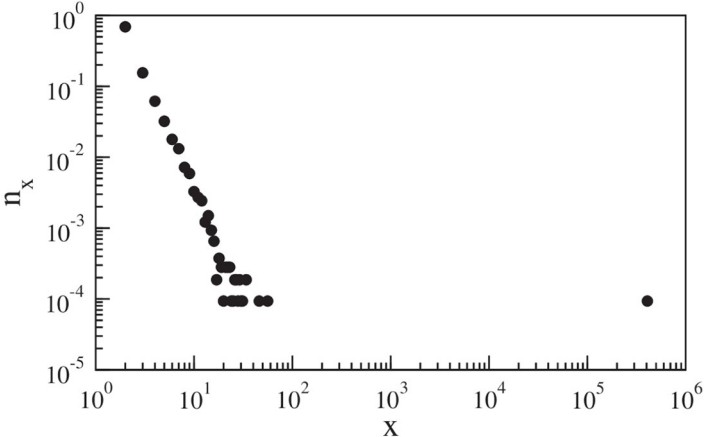

**Fig 3. Distribution $n_x$ of the component size $x$ in the network.** The largest weakly connected component contains ∼99% of the nodes in the entire network.

the total number of firms to the total number of the firms in the GWCC. The OUT component consists of the nodes from which the GSCC is reachable through directed paths downstream and is the largest, consisting of 41.1% of all firms. The GSCC (i.e., where any two nodes are reachable through directed paths), IN (i.e., nodes from which the GSCC is reachable through directed paths upstream), and TE (i.e., the rest of the nodes in the GWCC) are similar in size and comprise 16.4%, 22.3%, and 20.2% of all firms, respectively. For the Japanese supply chain network, the fraction of each component of the OUT, GSCC, IN, and TE is 26.2%, 49.7%, 20.6%, and 3.5%, respectively [18]. The GSCC in the Japanese supply chain network occupies half of the system, meaning that most firms are interconnected by small geodesic distances or the shortest-path lengths in the economy. This is a good contrast to the results for the global supply chain network that were observed in our study. However, by examining the shortest-path lengths from the GSCC to IN and OUT as shown in Table 4, one can observe that the upstream or downstream firms are located only a few steps away from the GSCC. This feature of the economic network is different from the bow-tie structure of many other complex networks [30]. In Japan, there is a large SCC in the ownership network [31]. This is well known as cross-shareholding, or "keiretsu" (a set of companies with interlocking business relationships and shareholdings) in Japanese. Correspondingly, there is a large SCC in the Japanese supply chain. On the other hand, the cross-shareholdings are not as pronounced in the world's firms as in Japan. As a result, the SCC is small in the global supply chain network.

## Flow characteristics: Community structure

Communities are detected in the largest weakly connected components of the network. We employ the map equation method [16] to discover the communities in the GWCC of the global

**Table 3. Bow-tie structure: The sizes of different components.**

| Component | #firms | Ratio (%) |
|---|---|---|
| GSCC | 66,798 | 16.4 |
| IN | 90,992 | 22.3 |
| OUT | 167,509 | 41.1 |
| TE | 82,228 | 20.2 |
| Total | 407,527 | 100 |

**Table 4. Shortest distances from the GSCC to IN/OUT.**

| IN to GSCC | | | OUT to GSCC | | |
|---|---|---|---|---|---|
| Distance | #firms | Ratio (%) | Distance | #firms | Ratio (%) |
| 1 | 82,761 | 90.954 | 1 | 153,755 | 91.789 |
| 2 | 7,430 | 8.165 | 2 | 11,885 | 7.095 |
| 3 | 665 | 0.731 | 3 | 1,582 | 0.944 |
| 4 | 104 | 0.114 | 4 | 250 | 0.149 |
| 5 | 17 | 0.019 | 5 | 26 | 0.015 |
| 6 | 10 | 0.011 | 6 | 10 | 0.006 |
| 7 | 5 | 0.005 | 7 | 1 | 0.001 |
| Total | 90, 992 | 100 | Total | 167, 509 | 100 |

supply chain network using the directed nature of the links. The directed links represent flow of goods from suppliers to customers. The detected communities are found in various sizes. The probability density distributions $D(s)$, of community sizes $s$ for an empirical network and its degree-preserving randomized network is shown in Fig 4(a). The distribution of the empirical network is more wider than it is for a randomized network. The maximum community size for the empirical distribution is $s_{max}$ = 1687 and in the randomized case it is $s_{max}$ = 551.

The bias in the direction of the flow between a pair of communities is measured by the polarization ratio, which is defined by $P_{ij} = |w_{ij} - w_{ji}|/(w_{ij} + w_{ji})$, where $w_{ij}$ is the total number of links from the $i$-th community to the $j$-th community. $P_{ij} = 1$ if the flow is totally biased from one community to the other and $P_{ij} = 0$ if the flow is evenly balanced between the communities. The total flow between a pair of communities is $L_{ij} = (w_{ij} + w_{ji})$. If we assume that there is no bias in the flow direction between any pair of communities, according to the null hypothesis, the values of $P_{ij}$ will fluctuates around zero with a standard deviation $\sigma = 1/\sqrt{L_{ij}}$. As can be seen in Fig 4(b), most of the values of the polarizability ratio $P_{ij}$ are significantly higher than the $2\sigma$ level, which is indicated by the dashed curve.

## Coarse-grained description: Bow-tie components and communities

We obtain a coarse-grained description of the global supply chain network in terms of its bow-tie and community structures. We examine the significant over-expression of different

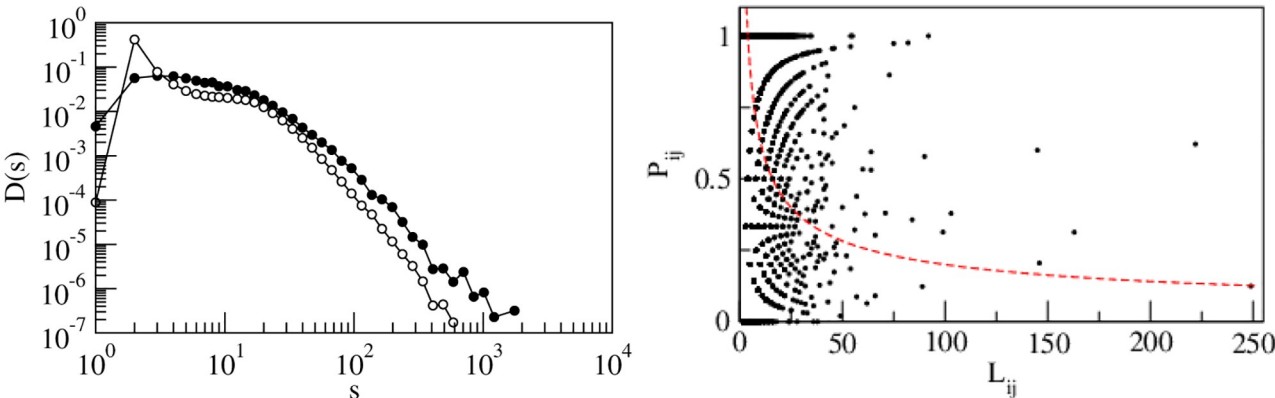

**Fig 4. Community sizes and polarizability.** (a) Distributions $D(s)$ of community sizes $s$ for the actual network with directed links (filled circles) and for its randomized counterpart (open circles). The distribution for the empirical network is wider than it is for the randomized network. (b) Polarizability of the direction of the links interconnecting communities.

**Table 5. Brief summary of the over-expression of sectors, countries, and bow-tie components in the 10 largest communities.**

| Rank | Size | Sector | Country | Bow-Tie Components |
|---|---|---|---|---|
| 1 | 1,687 | Consumer Discretionary (26.7%) | US (77.8%) | OUT (99.7%) |
| 2 | 1,632 | Consumer Discretionary (40.4%) | China (9.1%), UK (4.0%), France (3.7%), Germany (6.2%), Japan (9.1%), Malaysia (3.6%), New Zealand (1.3%) | IN (32.2%) |
| 3 | 1,179 | Consumer Discretionary (20.0%), Industrials (23.1%), Materials (15.8%) | China (12.4%), Japan (53.1%), Thailand (9.3%) | TE (99.6%) |
| 4 | 1,027 | Communication Services (9.4%) | Cambodia (0.6%), Indonesia (66.7%), Singapore (3.9%) | GSCC (20.3%), IN (55.1%) |
| 5 | 968 | Financials (62.1%) | Bahrain(0.7%), Bangladesh (1.8%), Hong Kong (5.0%), Hungary (1.0%), Italy (4.0%), Nepal (1.0%), Pakistan (14.0%), Singapore (3.8%), UAE (2.4%), Yemen (0.4%) | GSCC (22.7%), IN (63.1%) |
| 6 | 933 | Industrials (59.1%) | US (93.3%) | IN (82.3%) |
| 7 | 867 | Information Technology (50.2%) | Korea South (3.1%), Singapore (4.4%), Taiwan (10.4%) | GSCC (23.9%), IN (29.6%) |
| 8 | 824 | Consumer Discretionary (20.0%), Industrials (38.4%) | US (52.4%) | GSCC (29.5%), IN (28.9%) |
| 9 | 723 | Energy (12.3%), Industrials (17.6%), Materials (8.4%), Utilities (7.5%) | India(62.8%) | GSCC (28.5%) |
| 10 | 711 | - | Botswana (1%), South Africa (69.5%) | OUT (98.5%) |

attributes such as primary industry, sectors, firm location, and bow-tie components within communities. We demonstrated the detailed over-expression results in the 10 largest communities in Table 5. "Fraction" represents the percentage of firms in that community having a particular attribute. For example, in the first row and third column of the table, it represents 26.7% of firms in the largest community belong to the consumer discretionary sector. A variety of interesting features can be observed from the results of attribute over-expression. The largest community is comprised of the consumer discretionary sector based in the United State (US). Further analysis shows that these are private firms mainly from the automotive retail sector, which belong to the OUT component in the bow-tie structure of the global supply chain network. In the second largest community, we observe that the consumer discretionary sector is based in China, the United Kingdom (UK), France, Germany, Japan, Malaysia, and New Zealand. These firms belong to the IN component of the bow-tie structure. The firms of the third largest communities are from the consumer discretionary, industrials, and materials sectors, which are mainly based in Japan, China, and Thailand. These firms mostly belong to the TE component of the bow-tie structure.

We show the frequency of over-expression of the different components within the communities in the bow-tie structure in Fig 5. Here, we selected communities whose community size equals at least 10 firms. G-I indicates that both the GSCC and IN components were over-expressed in the communities. Similarly, G-O, G-T, I-O, I-T, and O-T represent over-expressions of GSCC-OUT, GSCC-TE, IN-OUT, IN-TE, and OUT-TE, respectively. This reflects the fact that most of the communities are solely composed of a particular component of the bow-tie structure. We also observe that there is a reasonable number of communities comprised of the combination of the GSCC and IN (i.e., the G-I component), which is also observed in the Japanese supply chain network [18]. This indicates that the flow of goods in the supply chain network is often confined within the GSCC and IN components as compared to any other combination of components of the bow-tie structure. Surprisingly, a large fraction of communities is in the TE component, not only supply but also procure any products and services from GSCC components.

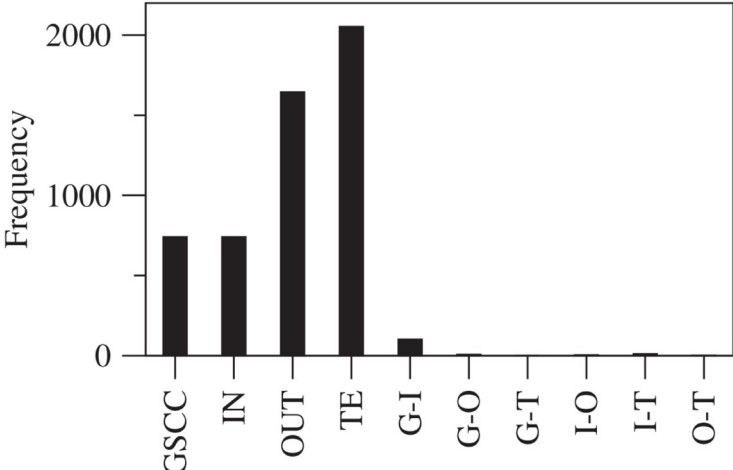

**Fig 5. Frequency of over-expression of bow-tie component within communities with a size of at least 10.** G-I indicates that both the GSCC and IN components are over-expressed in the communities. Similarly, G-O, G-T, I-O, I-T, and O-T represent over-expressions of GSCC-OUT, GSCC-TE, IN-OUT, IN-TE, and OUT-TE, respectively.

To visualize the co-occurrences of node attributes such as firms location and primary industry within the communities, we construct a network of over-expressions as described below. We construct a weighted and undirected network of countries from their over-expression in communities with sizes larger than 100 to show the interrelationships between countries. A link of weight one is placed between two countries if they over-express simultaneously within a community. Furthermore, we visualize the community structure of this network, as shown in Fig 6. Here, communities are detected using the modularity maximization technique. This demonstrates that each community is formed by geographically closely located countries. The first community consists mainly of South America, the second of Eastern and Southeastern Asia and Oceania, the third of Western and Northern Europe, the fourth of Western Asia, the fifth of Eastern Europe, the sixth of North America, and the seventh of Northern Africa. This result reflects the trade relations among the countries. The countries that belong to the same community have significantly stronger ties of trade relations between them.

Similarly, we also constructed a weighted undirected network of over-expressed primary industries, where a link of weight $w$ is present between two primary industries if they are over-expressed simultaneously in $w$ communities. As can be seen in Fig 7 below and S1 Fig in S1 Appendix, the clusters among the primary industries are formed based on their sector classifications. Here, communities are detected using the modularity maximization technique. The first community consists mainly of the industrial sector, the second of the consumer discretionary sector, the third of the health care sector, the fourth of the financials sector, the fifth of the energy and utility sectors, the sixth of the materials and industrials sector, the seventh of the information technology sector, and the eighth of the communication services. The result indicates the interactions of the flow of goods between the primary industries. Primary industries that belong to the same community depend heavily on each other.

## Verification of the efficient supply chain propositions

E. J.S. Hearnshaw *et al.* [9] studied the supply chain networks using complex network approach and proposed the following nine propositions. In this section, we investigate the

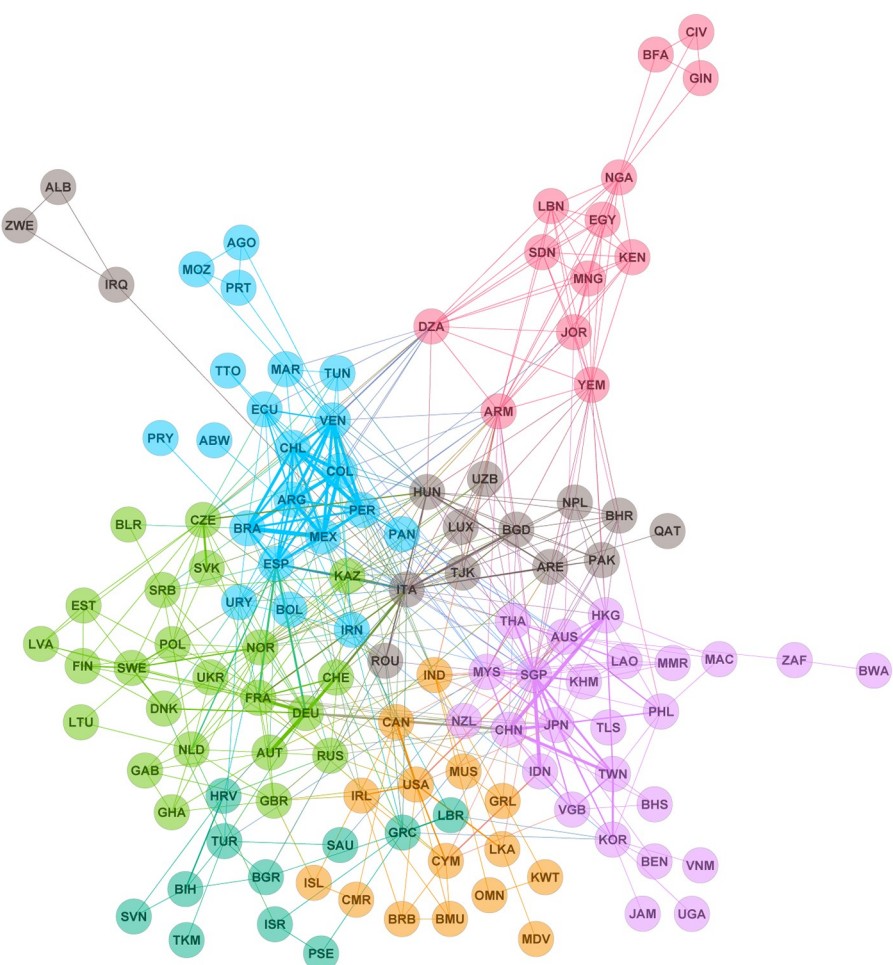

**Fig 6. Over-expression network of countries.** Different node colors indicate different communities of the network. Here, communities are detected using the modularity maximization technique.

validity of thees nine propositions according to the obtained results regarding the topological properties of the global supply chain network.

**Proposition S1.** S1: *Efficient supply chain systems demonstrate a short characteristic path length.*

The average path length of the global supply chain in the GWCC was found to be 5.370. The average path length in the small-world network $L_s$ is similar to the average path length in the random graph $Ls \sim L_r$. The average path length in the random graph $L_r$ is calculated by $L_r = \log N / \log <k> = 6.77$. Here, the number of nodes in the GWCC is $N = 407527$ and the average degree is $<k> = (<k_{in}> + <k_{out}>)/2 = 6.74$. This assumes that the degree distributions to be power-law distributions in the entire range of the degree with $\gamma_{in} = 2.42$ and $\gamma_{out} = 2.11$. The average in-degree and average out-degree are calculated by $<k_{in}> = k_{in}^{min}(\gamma_{in} - 1)/(\gamma_{in} - 2) = 3.38$, and $<k_{out}> = k_{out}^{min}(\gamma_{out} - 1)/(\gamma_{out} - 2) = 10.0$, where $k_{in}^{min} = 1$ and $k_{out}^{min} = 1$. The estimated value of $L_r = 6.77$ is close to the observed value 5.370. This reflects the small-world nature of the global supply chain network. Therefore, the estimation of the average path length validates Proposition S1 Original propositions of Hearnshaw S1 just mentioned a short characteristics path length. As described in the discussion of Table 4, the upstream or downstream firms are located only a few steps away from the GSCC.

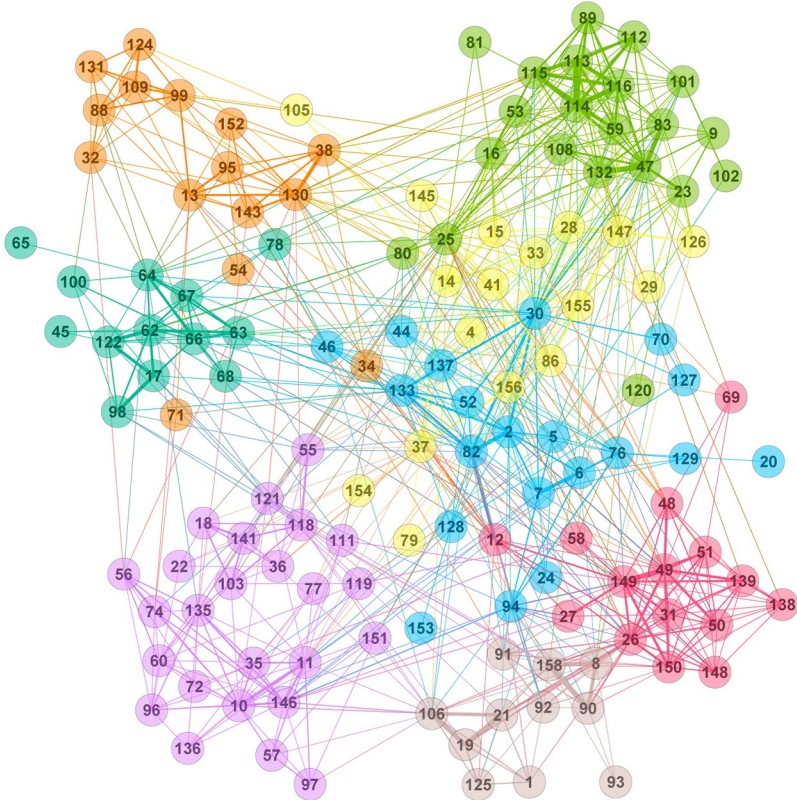

**Fig 7. Over-expression network of primary industries.** Different node colors indicate different communities of the network. Here, communities are detected using the modularity maximization technique. The IDs of the nodes are given in S3 Table in S1 Appendix.

The short path from the GSCC to IN and OUT was only discovered by analyzing the bow-tie structure. Therefore, the short path length in the bow-tie structure is a conceptual addition to the original propositions of Hearnshaw.

**Propositions S2.** S2: *The nodal degree distribution of efficient supply chain systems follows a power law, as indicated by the presence of hub firms.*

We observe that the probability density distributions for both nodal in and out degree have a heavy-tail nature in which the tail of the distribution is characterized by a power law of the form $P(k_{in/out}) \sim k^{-\gamma_{in/out}}$ with $\gamma_{in} = 2.42$ and $\gamma_{out} = 2.11$, respectively, as shown in Fig 2(a) and 2(b). The network in which the degree distribution is characterized by a power law includes hub firms. These hub firms are channel leader firms that control performance and provide system-wide coordination of the supply chain [32, 33]. The channel leader firms can exert their influence and provide opportunities and motivation for other firms to align themselves with their specific objectives [34]. The power law distributions are characterized by $\gamma_{in} = 2.42$ and $\gamma_{out} = 2.11$, which validates Proposition S2.

**Propositions S3.** S3: *Efficient supply chain systems demonstrate a high clustering coefficient.*

The clustering coefficient, a measure of three-point correlation, reveals cliquishness among the neighbors of a node. For most of the real-world network, the average clustering coefficient is a decaying function of degree having a form $\langle C(k) \rangle \sim k^{-\beta_k}$ with $\beta_k \leq 1.0$. We observe that the clustering coefficient in the supply chain network decays with an exponent $\beta_k = 0.46$ as

shown in Fig 2(c), which indicates the presence of a hierarchical structure. The clustering coefficient is clearly higher than the one observed in the randomized network, as shown in Fig 2(c). However, the value of the clustering coefficient is equivalent to many other networks, such as the power grid, mobile phone calls, science collaboration, etc. The observed moderate clustering coefficient indicates that Proposition S3:*It has a high clustering coefficient.* is valid.

**Propositions S4.**   S4: *The growth of efficient supply chain systems follows "fit-gets-richer" mechanism.*

It has been argued in the literature that such heavy-tail distributions of nodal degrees arise due to the "rich-get-richer" mechanism [27, 28]. The "rich-get-richer" principle means that large firms have more customers and suppliers than small firms. Preferential attachment in the "rich-get-richer" mechanism assumes that the acquisition of new links by a firm is determined solely by the number of its existing links. This assumption leads to the number of links being proportional to their duration in the supply chain. However, one can often observe that older firms are outstripped by new entrant firms. There is a need, therefore, to include the "fitness" of the firms to account for new entrants that can quickly dominate supply chains. By introducing the "fit-gets-richer" mechanism [35], the fitter nodes have a greater acquisition rate for links and, therefore, the resulting network possess a scale-free property. The heavy-tail distribution of nodal degrees and the overtaking of older firms by new entrant firms validates Proposition S4.

**Propositions S5.**   S5: *The power law degree distribution of efficient supply chain system is truncated.*

The power law distributions $P(k_{in/out}) \sim k^{-\gamma_{in/out}}$ with $\gamma_{in} = 2.42$ and $\gamma_{out} = 2.11$ respectively are observed in the middle region of the distributions as shown in Fig 2(a) and 2(b). The tail region of both distributions seem like truncated due to a finite system size. Especially this tendency is evident for $P(k_{out})$. This phenomenon is said to be caused by four reasons [9]. First, the finite size of marketplaces generates a truncated power law degree distribution. Second, there are practical reasons in the operation of firms that limit the ability of firms to indefinitely form and maintain exchange relationships. Third, when new links are to be formed with a hub firms, incomplete information generates uncertainty which might costs higher than transaction costs. If these costs are unacceptable, the firms will scrap the deal with the hub firms. Finally, the aging and depreciation of firms limits their growth. The observed truncation or cutoff in the tail region of the degree distribution validates Proposition S5.

**Propositions S7.**   S7: *Efficient supply chain systems demonstrate a pronounced community structure with overlapping boundaries.*

We employ the map equation method [16] to uncover the communities in the GWCC of the global supply chain network. The detected communities were found in various sizes. The probability density distributions $D(s)$ of community size $s$ for the empirical network and its degree-preserving randomized network are shown in Fig 4(a). The distribution of the empirical network is wider than that of the randomized network. In Table 5, the over-expression of sectors and countries in the 10 largest communities is shown. Communities in a supply chain are bound together in clusters predominantly connected by horizontal relationships between firms with similar interests and functions. However, we empirically observed that all firms within a community are not entirely cooperative, as shown in Table 5. The over-expression of countries and sectors in the large communities shown in Table 5 allows us to characterize community formation by two factors. One is the over-expression of countries in the large communities shown in Fig 6. The other is the over-expression network of sectors shown in Fig 7. In these figures, a country tends to form communities in neighboring countries, and a sector tends to form communities in the same industry. This result suggests that community

formation is due to multiple factors, and the over-expression networks in Figs 6 and 7 provide support for the appropriateness of the overlapping boundary of community formation in Proposition S7. Therefore, community formation in the supply chain possesses overlapping boundaries. These results validates Proposition S7.

**Remaining propositions.** The supply chain data have no weight on links. Therefore, the two propositions: Proposition S6: *The link weight distribution of efficient supply chain systems follows a power law* and Proposition S9: *Resilient supply chain systems demonstrate a power law distribution for link-weights* are not applicable to the analysis in this paper. In addition, we concentrated on the topological properties of the supply chain network and, therefore, the resilience of the system, so Proposition S8: *The fitness of hub firms determines the resilience of supply chain systems against both random disturbances and targeted attacks* is also out of the scope of the current study.

## Conclusions

In this paper, we studied the topological properties of the global supply chain network to empirically test the efficient supply chain propositions proposed by E. J.S. Hearnshaw *et al*. We verified the propositions by conducting a large-scale empirical analysis of global supply chain data. The topological properties of the global supply chain network are empirically studied from three different angles: node-level characteristics, mesoscopic structural characteristics, and flow characteristics. An analysis from these three different angles was required to test the efficient supply chain propositions and obtain a coarse-grained description of the global supply chain network in terms of its bow-tie and community structures.

The global supply chain data in the year 2017 were constructed by collecting various company data from Standard & Poor's Capital IQ platform. The total number of firms and directed links in our data were 437, 453 and 948, 247, respectively.

Degree distribution is characterized by a power law of the form with $\gamma_{in}$ = 2.42 and $\gamma_{out}$ = 2.11. The clustering coefficient decays $\langle C(k) \rangle \sim k^{-\beta_k}$ with an exponent $\beta_k$ = 0.46. This indicates the presence of a hierarchical structure to the supply chain network. We observed that $\langle k_{nn}(k) \rangle$ does not depend on $k$ and remains more or less constant with $k$, indicating the absence of nodal degree-degree correlation.

The bow-tie structure of the GWCC revealed that the OUT component was the largest and consisted of 41.1% of all firms. The GSCC component comprised 16.4% of all firms. We observed that upstream or downstream firms were located a few steps away from the GSCC. Then, we discovered the community structure of the network using the map equation method and characterized them according to their locations and industry classifications. We observed that the largest community was comprised of private firms mainly from the automotive retail sector based in the US. These firms are belong to the OUT component in the bow-tie structure of the global supply chain network. This indicates that the retail firms generally belong to the OUT component of the bow-ties structure.

Finally, we tested the validity of the nine propositions on the supply chain network based on the results obtained from the topological properties. We confirmed the validity of Propositions S1 (i.e., short path length), Proposition S2 (i.e., power-law degree distribution), Proposition S3 (i.e., high clustering coefficient), Proposition S4 (i.e., "fit-gets-richer" growth mechanism), Proposition S5 (i.e., truncation of power-law degree distribution), and Proposition S7 (i.e., community structure with overlapping boundaries) regarding the global supply chain network. While the original propositions S1 just mentioned a short path length, we found the short path from the GSCC to IN and OUT by analyzing the bow-tie structure. Therefore, the short path length in the bow-tie structure is a conceptual addition to the original

propositions of Hearnshaw. However, the propositions related to link weight and the resilient nature of the network were not confirmed due to the limitations of our data and the narrow scope of the current study. Such analysis will be left for a future study.

## Supporting information

**S1 Appendix.**
(PDF)

## Acknowledgments

We are grateful to Y. Fujiwara, H. Aoyama, H. Iyetomi, W. Souma, and H. Yoshikawa for their insightful comments and encouragement.

## Author Contributions

**Conceptualization:** Yuichi Ikeda.

**Methodology:** Abhijit Chakraborty, Yuichi Ikeda.

**Project administration:** Yuichi Ikeda.

**Software:** Abhijit Chakraborty.

**Supervision:** Yuichi Ikeda.

**Visualization:** Abhijit Chakraborty.

**Writing – original draft:** Abhijit Chakraborty, Yuichi Ikeda.

**Writing – review & editing:** Abhijit Chakraborty, Yuichi Ikeda.

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
