## [Decision Letter · Decision Letter 0]

16 Apr 2020

PONE-D-20-05766

Bow-tie structure and community identification of global supply chain network

PLOS ONE

Dear Professor Ikeda,

Thank you for submitting your manuscript to PLOS ONE. After careful consideration, we feel that it has merit but does not fully meet PLOS ONE’s publication criteria as it currently stands. Therefore, we invite you to submit a revised version of the manuscript that addresses the points raised during the review process.

We would appreciate receiving your revised manuscript by May 31 2020 11:59PM. To enhance the reproducibility of your results, we recommend that if applicable you deposit your laboratory protocols in protocols.io, where a protocol can be assigned its own identifier (DOI) such that it can be cited independently in the future. For instructions see: http://journals.plos.org/plosone/s/submission-guidelines#loc-laboratory-protocols

We look forward to receiving your revised manuscript.

Kind regards,

Hocine Cherifi

Academic Editor

PLOS ONE

Reviewers' comments:

Reviewer's Responses to Questions

**Comments to the Author**

1. Is the manuscript technically sound, and do the data support the conclusions?

Reviewer #1: Yes

Reviewer #2: Partly

Reviewer #3: Partly

2. Has the statistical analysis been performed appropriately and rigorously? 

Reviewer #1: Yes

Reviewer #2: No

Reviewer #3: Yes

3. Have the authors made all data underlying the findings in their manuscript fully available?

Reviewer #1: Yes

Reviewer #2: No

Reviewer #3: No

4. Is the manuscript presented in an intelligible fashion and written in standard English?

Reviewer #1: No

Reviewer #2: Yes

Reviewer #3: Yes

5. Review Comments to the Author

Reviewer #1: The paper investigates the topological properties of a sample from the global supply chain network and by doing so, it empirically test some propositions inherited from earlier research in the literature. Although the topic is of clear interest and the paper applies adequate methodology to address the research questions, there are several drawbacks found mainly in the exposition of the methodology/results which call for major revisions in the manuscript.

First, I would like to add two general comments on the paper.

1. It turns out by the end of the paper, that the main purpose in the background is to empirically test the propositions given in Hearnshaw et. al (2013) in IJOPM. Although this is marked in the introduction, the Abstract does not indicate this and the structure of the paper suggest a different setup. The reader is uncertain about why those topological properties are analyzed which are in the paper until it becomes clear at the end that because these provide the basis for testing the propositions. When discussing the empirical tests, the text continuously refer back to different parts of the topological analysis. I would feel the paper more coherent if this dichotomy between the topological analysis and the tests were resolved. I suggest to merge the two parts and discuss the relevant topological properties wherever they contribute to testing a given proposition. At the same time, it would be important to make very clear from the beginning that the paper uses the topological analysis to give empirical support for the mentioned propositions.

2. The English of the paper is to be significantly improved – without being a native speaker, I felt the text quite difficult to read sometimes and in many places obvious grammatical mistakes are left.

Then, some specific comments are made in what follows.

3. Line 7: there should be more discussion of what ‘collective motions’ the Authors are talking about.

4. In the beginning of the Data section, there should be clearly indicated how much companies are involved in the sample, what is their industrial composition, what is the type of the companies (are they large, listed companies or does the sample include SME-s), according to what principle did the companies got into the sample, what is the time period for which the data is relevant? More generally, to what extent can we consider the sample as representative – a crucial issue with respect to the significance and interpretation of the results. Also, we do not know how representative the links are in the sample: according to what principles do links show up in the sample? All links are accounted for, just large transactions, etc.? The representativity of the sample largely determines how meaningful the results are: for example detected communities may be trivial if the sample has some selection bias.

5. The numbers in line 107 do not match with those in Table 1 and Table 2.

6. In line 111: how do these statistics prove the goodness of the data?

7. The number of firms in Table 2 is a magnitude larger than the number reported in Table S2. Maybe this is the number of links gain, but then why is it different from the link number in Table 1?

8. The paper always refer to supply chain networks, but the data covers many other types of links. However, it is not clear if the Authors use only supplier links from the data or other links as well? In the latter case, referring to supply chains is not adequate.

9. In the Community detection section the authors talk extensively about the modularity optimization method, which is not used in the paper. The first paragraph then is unnecessary, should be compressed into a sentence mentioning this method and the drawbacks (plus: why not mentioning other methods as well?).

10. Line 141: the Authors argue that map equation is the best performing method. But according to what? Accuracy or time? Or something else?

11. In line 216 the Authors draw attention to a difference between their results and previous ones. There should be a discussion of this difference.

12. Similarly, in line 220 there is a difference revealed with respect to many other networks. There should be a discussion about the reasons of this difference.

13. In Figures 2 and 4 the random network shows up as a reference. The references represent averages from many randomized networks. However, deviation from this average does not indicate that the observed properties significantly differ from that obtained in a random world. There should be a confidence interval drawn around the average in order to see if this is really the case. The same issue can be raised with respect to the characteristic path length around line 296.

14. The paper uses only one method for community detection. The question naturally arises if the results of community detection are robust to the choice of (adequate) methods?

15. It is not evident what the two representations of the supply chain network in Figures 5 and 6 tell us. How they can be interpreted? Why they are important to be analyzed?

16. What do the ratios in column 3 of Table 5 represent?

17. In line 319 the Authors say that clustering is moderate. But compared to what?

18. In the discussion of Proposition S5, the Authors claim that the degree distributions are exponentially cut off at the tails. In order to convincingly prove this, they should either supply some statistical measurements or at least draw the red lines in Figure 2 up to the edge of the graphs to see the deviation from power law.

Reviewer #2: This paper provides an empirical analysis of topological properties of global supply chain network in terms of degree distribution, hierarchical structure, and degree-degree correlation in the global supply

chain network.

I have many concerns about this paper. The analysis sounds to me quite standard and based on the application of standard techniques in networks theory. From my point of view the contribution to the literature is very limited.

Additionally I found the description of the analysis a bit limited. I summarize just a few examples:

- a clustering coefficient has been computed. Several clustering coefficients exist in the literature and different versions have been provided for directed networks able to identify different triangles (as in clustering, out clustering, cycles, etc). It is not clear which kind of clustering has been used and how

- the author/authors affirm that the distribution follows a power law by means of the observation of the figure. A statistical test is typically needed to provide that a distribution assures a good fit to the data.

- it's not clear why the connected component is studied only in the undirected case

Finally, the discussion of the proposition seems to me quite standard and I do not see a so significant contribution to the literature.

Reviewer #3: Review of PONE-D-20-05766 “Bow-tie structure and community identification of global supply chain network “

This manuscript analyzes a large supply chain network with 1,417,169 firms globally. The key aspect of the data is a directed graph of supplier relations, with 1,439,101 links. The manuscript demonstrates a bow-tie structure, and addresses nine claims, expectations about efficient supply chain networks.

While the analysis is rigorous and the data work is careful and appropriate, the conceptualization of the manuscript is insufficient. The motivations to demonstrate a bow-tie structure, and the consequences of a bow tie structure to economic outcomes is not clear. The nine claims about efficient supply chain networks are analyzed without addressing data on efficiency. The two tasks accomplished in the paper – demonstrating a bow-tie structure and addressing nine claims about efficient supply chain networks – are mixed together without clarifying the relationship between these two tasks. Should we think of a bow-tie structure within the framework of an efficient supply chain network? How do the nine claims relate to the bow tie structure?

Many figures are un-related to the argument, and are only passingly mentioned in the main text (Fig. 5-7). The network figures are not styled consistently, and the edges are not clearly visible.

In general, this manuscript would need to clarify its goals much better in the introduction, and the empirical results would need to be interpreted more carefully. Results and various goals of the manuscript should be much better integrated into a coherent argument.

6. PLOS authors have the option to publish the peer review history of their article (what does this mean?). If published, this will include your full peer review and any attached files.

Reviewer #1: No

Reviewer #2: No

Reviewer #3: No

---

## [Author Response · Author response to Decision Letter 0]

26 Jun 2020

View Letter

Date: Apr 16 2020 10:23AM

To: "Yuichi Ikeda" ikeda.yuichi.2w@kyoto-u.ac.jp

From: "PLOS ONE" plosone@plos.org

Subject: PLOS ONE Decision: Revision required [PONE-D-20-05766]

PONE-D-20-05766

Bow-tie structure and community identification of global supply chain network

PLOS ONE

We thank all the reviewers for giving us invaluable comments. We consider all the comments seriously and made careful reexaminations. We made the following changes shown with underline and red color for each comment.

Reviewer #1: The paper investigates the topological properties of a sample from the global supply chain network and by doing so, it empirically test some propositions inherited from earlier research in the literature. Although the topic is of clear interest and the paper applies adequate methodology to address the research questions, there are several drawbacks found mainly in the exposition of the methodology/results which call for major revisions in the manuscript.

First, I would like to add two general comments on the paper.

#1-1. It turns out by the end of the paper, that the main purpose in the background is to empirically test the propositions given in Hearnshaw et. al (2013) in IJOPM. Although this is marked in the introduction, the Abstract does not indicate this and the structure of the paper suggest a different setup. The reader is uncertain about why those topological properties are analyzed which are in the paper until it becomes clear at the end that because these provide the basis for testing the propositions. When discussing the empirical tests, the text continuously refer back to different parts of the topological analysis. I would feel the paper more coherent if this dichotomy between the topological analysis and the tests were resolved. I suggest to merge the two parts and discuss the relevant topological properties wherever they contribute to testing a given proposition. At the same time, it would be important to make very clear from the beginning that the paper uses the topological analysis to give empirical support for the mentioned propositions.

-> 

1. We changed the first sentence of the abstract as follows. 

“In this paper, we study the topological properties of the global supply chain network in terms of its degree distribution, clustering coefficient, degree-degree correlation, bow-tie structure, and community structure to verify the efficient supply chain propositions proposed by E. J.S. Hearnshaw et al..”

2. We added the following description in section “Introduction”. (line 79 in the revised manuscript)

“In this paper, we focus on the topological properties of the global supply chain network to empirically verify the efficient supply chain propositions put forth by E.~J.S.~Hearnshaw~{\\em et al.}. We contribute to the literature a verification of these propositions by conducting a large-scale empirical analysis of global supply chain data. This study on the topological properties of the global supply chain network is a first step in understanding the globalized world economy under a microscopic view. We study degree distribution, clustering coefficient, and degree-degree correlation in the global supply chain network. A bow-tie structure was uncovered based on the supply of goods and services along directed links. Then, we uncovered the community structures of the network using the map equation method and characterized them according to their locations and industry classifications. Furthermore, the composition of the communities in terms of bow-tie components is analyzed. Finally, we investigate the validity of E.~J.S.~Hearnshaw~{\\em et al.}'s nine propositions regarding the efficiency of supply chain networks \\cite{Hearnshaw2013} based on the results obtained regarding the topological properties of the global supply chain network.”

3. We’d like to ask the editor allow us to change the title of the paper to show our goal clearly from the beginning to respond the reviewers comment. The new title is “Verifying “efficient supply chain propositions” using topological characterization of the global supply chain network”.

#1-2. The English of the paper is to be significantly improved – without being a native speaker, I felt the text quite difficult to read sometimes and in many places obvious grammatical mistakes are left.

-> After finishing the revision, I used English proofreading service. I hope that the English used in the revised version is sufficiently improved. 

Then, some specific comments are made in what follows.

#1-3. Line 7: there should be more discussion of what ‘collective motions’ the Authors are talking about.

-> We added the following sentence to explain the meaning of ‘collective motions’ in section “Introduction”. (line 7 in the revised manuscript) 

“A variety of collective motions exist in natural and social phenomena. Some examples of the collective motions of the global economy include the synchronization of the business cycle, global economic crises, and chain bankruptcies.”

#1-4. In the beginning of the Data section, there should be clearly indicated how much companies are involved in the sample, what is their industrial composition, what is the type of the companies (are they large, listed companies or does the sample include SME-s), according to what principle did the companies got into the sample, what is the time period for which the data is relevant? More generally, to what extent can we consider the sample as representative – a crucial issue with respect to the significance and interpretation of the results. Also, we do not know how representative the links are in the sample: according to what principles do links show up in the sample? All links are accounted for, just large transactions, etc.? The representativity of the sample largely determines how meaningful the results are: for example detected communities may be trivial if the sample has some selection bias.

-> We changed the description in section “data” as follows. (line 120 in the revised manuscript)

“In Table 1, the types of business relationships of all firms are summarized. We note that the links in the dataset are dominated by supply chain business relationships. In Table 2, the supplier types of all firms are summarized. We note that suppliers are dominated by private and public firms. Therefore, the characteristics of the dataset reflect the nature of the global supply chain network. 

We preprocessed the raw data in the following way. We removed all creditor relationships, isolated nodes, parallel links, and self-loops from the dataset being used in our analysis. The total number of firms and directed links used in our analysis are $437,453$ and $948,247$, respectively. The number of firms and total revenue of firms in each country are listed in Table S1 in Appendix S1. Firm distributions for the different sectors are listed in Table S2 in Appendix S1. It is noted that Table S2 represents the data after preprocessing, while the Table 2 represents the raw data. Hereafter, the preprocessed data is referred to as the global supply chain data in the year 2017.

Aggregated revenue is compared to the gross domestic product (GDP) of each country, as shown in Fig 1. These statistics provide evidence for the goodness of data coverage of our global supply chain data. GDP data were collected from \\url{https://data.worldbank.org/} and are therefore in the public domain.”

#1-5. The numbers in line 107 do not match with those in Table 1 and Table 2.

-> We have removed all the creditor relationships, isolated nodes, parallel links and self loops for our analysis. Please see the response to Comment #1-4. (line 125 in the revised manuscript)

“We preprocessed the raw data in the following way. We removed all creditor relationships, isolated nodes, parallel links, and self-loops from the dataset being used in our analysis. The total number of firms and directed links used in our analysis are $437,453$ and $948,247$, respectively. The number of firms and total revenue of firms in each country are listed in Table S1 in Appendix S1. Firm distributions for the different sectors are listed in Table S2 in Appendix S1. It is noted that Table S2 represents the data after preprocessing, while the Table 2 represents the raw data. Hereafter, the preprocessed data is referred to as the global supply chain data in the year 2017.”

#1-6. In line 111: how do these statistics prove the goodness of the data?

-> We corrected the axis level and changed the caption: The red line represents best power law fit to the data of the form Revenue = 2.81 X10^{-5} GDP^{1.369} . Please see Fig.1 and the response to Comment 4. (line 133 in the revised manuscript)

 “Aggregated revenue is compared to the gross domestic product (GDP) of each country, as shown in Fig 1. These statistics provide evidence for the goodness of data coverage of our global supply chain data. GDP data were collected from \\url{https://data.worldbank.org/} and are therefore in the public domain.”

#1-7. The number of firms in Table 2 is a magnitude larger than the number reported in Table S2. Maybe this is the number of links gain, but then why is it different from the link number in Table 1?

-> The table 2 represents the raw data. We have preprocessed the raw data in the following way. We have removed all the creditor relationships, isolated nodes, parallel links and self loops from the raw data. Please see the response to Comment #1-4. Table S2 represents the data after preprocessing that we have used for our analysis.

#1-8. The paper always refer to supply chain networks, but the data covers many other types of links. However, it is not clear if the Authors use only supplier links from the data or other links as well? In the latter case, referring to supply chains is not adequate.

-> We have removed creditor relationships from the raw data. The data mainly represents the supplier relationships. Please see the response to Comment #1-4.

#1-9. In the Community detection section the authors talk extensively about the modularity optimization method, which is not used in the paper. The first paragraph then is unnecessary, should be compressed into a sentence mentioning this method and the drawbacks (plus: why not mentioning other methods as well?).

#1-10. Line 141: the Authors argue that map equation is the best performing method. But according to what? Accuracy or time? Or something else?

-> We changed the explanation of community analysis in section of Methods as follows.

“\\subsection*{Flow characteristics} (line 170 in the revised manuscript)

 Empirical networks are generally non-homogeneous with a high local link density. Community detection captures highly connected groups of nodes as modules. %It provides a coarse-grained description of very large scale networks. Modularity maximization [11] is one of the popular methods for detecting communities. With this method, one maximizes the modularity index. Modularity is defined as the fraction of intra-community links, subtracting the expected fraction, given a random distribution. However, this method suffers from a resolution limit problem [12] when applied to large networks. Despite this problem, the modularity maximization is still an effective method for detecting communities in moderate sized undirected and directed networks. We use the modularity maximization to detect communities of over-expression networks in our study.

 The map equation method [13] detects communities using the flow dynamics of random walkers on the network. We use the map equation method for our analysis of the global supply chain network since it is a directed network of suppliers and customers in which link represents the flow of goods. This method is one of the most suitable community detection techniques for detecting communities in a network [14] based on the flow of random walkers that does not suffer from the resolution limit because our main interest in community detection is in extracting flow characteristics.”

#1-11. In line 216 the Authors draw attention to a difference between their results and previous ones. There should be a discussion of this difference.

#1-12. Similarly, in line 220 there is a difference revealed with respect to many other networks. There should be a discussion about the reasons of this difference.

-> We thank the referee for pointing this out. Although the reasoning on these differences itself is of interest, it would be difficult to obtain the reasoning from the analysis of the current data.

#1-13. In Figures 2 and 4 the random network shows up as a reference. The references represent averages from many randomized networks. However, deviation from this average does not indicate that the observed properties significantly differ from that obtained in a random world. There should be a confidence interval drawn around the average in order to see if this is really the case. The same issue can be raised with respect to the characteristic path length around line 296.

-> We have added the standard deviation as vertical line (which are very small) in Fig 2 as the reviewer suggested. For Fig 4, we mainly observe that the community size distribution is wider than the randomized counterpart. The max community size for empirical distribution is s_max = 1687 and in the randomized case it is s_max= 551. The average is taken over 13 uncorrelated degree preserving randomize networks.

Please see the first paragraph of section “Flow characteristics: community structure” We added the following description. (line 276 in the revised manuscript)

“The distribution of the empirical network is more wider than it is for a randomized network. The maximum community size for the empirical distribution is s_max = 1687 and in the randomized case it is s_max = 551.”

#1-14. The paper uses only one method for community detection. The question naturally arises if the results of community detection are robust to the choice of (adequate) methods?

-> The map equation is the most suitable method for us. We elaborate the explanation in section “Methods” as follows. Please also see the response to Comment 9. (line 181 in the revised manuscript)

“The map equation method [13] detects communities using the flow dynamics of random walkers on the network. We use the map equation method for our analysis of the global supply chain network since it is a directed network of suppliers and customers in which link represents the flow of goods. This method is one of the most suitable community detection techniques for detecting communities in a network [14] based on the flow of random walkers that does not suffer from the resolution limit because our main interest in community detection is in extracting flow characteristics.”

#1-15. It is not evident what the two representations of the supply chain network in Figures 5 and 6 tell us. How they can be interpreted? Why they are important to be analyzed?

-> We made the follows three revisions in section “Coarse-grained description: bow-tie components and communities”.

Add at line 324: “To visualize the co-occurrence of node attributes such as firms' locations and their primary industries. within the communities, we construct a network of overexpressions as described below.”

Add at line 336: “This result reflects trade relations among the countries. The countries belong to the same community are having significantly stronger ties of trade relations between them.”

Add at line 347: “The result indicates the interactions in terms of flow of goods between the primary industries. The primary industries belonging to the same community depend strongly on each other.”

#1-16. What do the ratios in column 3 of Table 5 represent?

-> Fraction represents the percentage of firms in that community having the particular attribute.

Example: First row and column 3 of table 5: Consumer Discretionary (26.7%).

It represents 26.7 % of firms in the largest community belonging to the Consumer Discretionary sector.

We changed the description as follows. (line 295 in the revised manuscript)

“We demonstrated the detailed over-expression results in the 10 largest communities in Table 5. “Fraction” represents the percentage of firms in that community having a particular attribute. For example, in the first row and third column of the table, it represents 26.7 % of firms in the largest community belong to the consumer discretionary sector.”

#1-17. In line 319 the Authors say that clustering is moderate. But compared to what?

-> We thank the referee for pointing this out. It is compared with the randomized version of the network. Clearly, the clustering coefficient is higher than that observed in the randomized network [Fig 2(c)]. However, it is not too high (<<1). This is the reason we have used the term moderate. 

#1-18. In the discussion of Proposition S5, the Authors claim that the degree distributions are exponentially cut off at the tails. In order to convincingly prove this, they should either supply some statistical measurements or at least draw the red lines in Figure 2 up to the edge of the graphs to see the deviation from power law.

-< We added the following explanation in section “Node-level characteristics” (line 211 in the revised manuscript)

“It is, however, noted that we are not claiming that the entire distribution follows a power law distribution. We measured the exponent from the slope of the straight portion of the intermediate range of degrees. The tail region of both distributions seem to be truncated due to the finite system size.”

 

Reviewer #2: This paper provides an empirical analysis of topological properties of global supply chain network in terms of degree distribution, hierarchical structure, and degree-degree correlation in the global supply chain network.

#2-1. I have many concerns about this paper. The analysis sounds to me quite standard and based on the application of standard techniques in networks theory. From my point of view the contribution to the literature is very limited.

-> We added the following description in section “Introduction”. (line 79 in the revised manuscript)

“In this paper, we focus on the topological properties of the global supply chain network to empirically verify the efficient supply chain propositions put forth by E.~J.S.~Hearnshaw~{\\em et al.}. We contribute to the literature a verification of these propositions by conducting a large-scale empirical analysis of global supply chain data. This study on the topological properties of the global supply chain network is a first step in understanding the globalized world economy under a microscopic view. 

Additionally I found the description of the analysis a bit limited. I summarize just a few examples:

#2-2. a clustering coefficient has been computed. Several clustering coefficients exist in the literature and different versions have been provided for directed networks able to identify different triangles (as in clustering, out clustering, cycles, etc). It is not clear which kind of clustering has been used and how

-> We thank the referee for pointing this out. For the calculation of clustering coefficient we have considered a simple undirected version of the network. 

#2-3. the author/authors affirm that the distribution follows a power law by means of the observation of the figure. A statistical test is typically needed to provide that a distribution assures a good fit to the data.

-> We added the following explanation in section “Node-level characteristics” (line 211 in the revised manuscript). Please see the response to Comment #1-18.

“It is, however, noted that we are not claiming that the entire distribution follows a power law distribution. We measured the exponent from the slope of the straight portion of the intermediate range of degrees. The tail region of both distributions seem to be truncated due to the finite system size.”

#2-4. it's not clear why the connected component is studied only in the undirected case

-> Community detection methods are mainly applied on connected components of the network. If we use a fragmented network, it can give some trivial results. Moreover,the connected component represents ~ 99% of the nodes of the entire network.

#2-5. Finally, the discussion of the proposition seems to me quite standard and I do not see a so significant contribution to the literature.

-> In section “Introduction”, we added the following explanation to clarify our contribution to the literature. (line 62 in the revised manuscript) 

“In E.~J.S.~Hearnshaw~{\\em et al.}'s study, the term ``efficient supply chain” was used to characterize the fact that a variety of resources such as financial, human, technological, and physical resources were effectively used in supply chains. These nine propositions S1 through S9 are related to path length, power law degree distribution, clustering coefficients, preferential attachment growth mechanism, truncated power law connectivity distribution, power law distribution of node strength, community structure with overlapping boundaries, resilience against random failure and targeted attack, and core-periphery structure, respectively. They attempt to explain the various functions of a supply chain according to the structural characteristics of supply chain networks. It is, however, noted that the nine propositions were proposed by a theoretical study based on a survey of the existing literature using only small-scale empirical data. Its empirical verification is rather weak and therefore a large-scale empirical study using more comprehensive data would be highly desired and contribute significantly to the extant literature on supply chain networks.”

 

Reviewer #3: Review of PONE-D-20-05766 “Bow-tie structure and community identification of global supply chain network “

This manuscript analyzes a large supply chain network with 1,417,169 firms globally. The key aspect of the data is a directed graph of supplier relations, with 1,439,101 links. The manuscript demonstrates a bow-tie structure, and addresses nine claims, expectations about efficient supply chain networks.

#3-1. While the analysis is rigorous and the data work is careful and appropriate, the conceptualization of the manuscript is insufficient. 

-> 1. We added the following description in section “Methods of topological characterization”. (line 137 in the revised manuscript)

“In order to verify Hearnshaw et al.'s efficient supply chain propositions, the topological properties of the global supply chain network are empirically studied from three different angles: their node-level characteristics, mesoscopic structural characteristics, and flow characteristics. The analysis from these three different angles is required to verify the efficient supply chain propositions and obtain a coarse-grained description of the global supply chain network in terms of bow-tie and community structures. ”

2. We restructured section "Methods of topological characterization" (line 137). The title of three subsections are listed in the following. 

subsection "Node-level characteristics (line 144)

subsection "Mesoscopic structural characteristics (line 156)

subsection "Flow characteristics (line 170)

subsection "Coarse-grained description (line 193)

3. We restructured section "Results” (line 206). The title of three subsections are listed in the following. 

subsection "Node-level characteristics (line 207)

subsection "Mesoscopic structural characteristics: bow-tie structure (line 252)

subsection "Flow characteristics: community structure (line 271)

subsection "Coarse-grained description: bow-tie components and communities (line 291)

4. Please see the response to Comment #1-1 for the related revisions.

#3-2. The motivations to demonstrate a bow-tie structure, and the consequences of a bow tie structure to economic outcomes is not clear. 

-> We elaborated the motivations to demonstrate a bow-tie structure in section “Methods of topological characterization” as follows. (line 137 in the revised manuscript)

“In order to verify Hearnshaw~{\\em et al.}'s efficient supply chain propositions,

the topological properties of the global supply chain network are empirically studied

from three different angles: their node-level characteristics, mesoscopic structural characteristics, and flow characteristics. The analysis from these three different angles is required to verify the efficient supply chain propositions and obtain a coarse-grained description of the global supply chain network in terms of bow-tie and community structures. ”

-> We elaborated the consequences of a bow tie structure to economic outcomes in section “Coarse-grained description: bow-tie components and communities” as follows. (line 290 in the revised manuscript)

“We obtain a coarse-grained description of the global supply chain network in terms of its bow-tie and community structures. We examine the significant over-expression of different attributes such as primary industry, sectors, firm location, and bow-tie components within communities. We demonstrated the detailed over-expression results in the $10$ largest communities in Table 5. ``Fraction" represents the percentage of firms in that community having a particular attribute. For example, in the first row and third column of the table, it represents 26.7 \\% of firms in the largest community belong to the consumer discretionary sector. A variety of interesting features can be observed from the results of attribute over-expression. The largest community is comprised of the consumer discretionary sector based in the United State (US). Further analysis shows that these are private firms mainly from the automotive retail sector, which belong to the OUT component in the bow-tie structure of the global supply chain network. In the second largest community, we observe that the consumer discretionary sector is based in China, the United Kingdom (UK), France, Germany, Japan, Malaysia, and New Zealand. These firms belong to the IN component of the bow-tie structure. 

The firms of the third largest communities are from the consumer discretionary, industrials, and materials sectors, which are mainly based in Japan, China, and Thailand. These firms mostly belong to the TE component of the bow-tie structure. 

We show the frequency of over-expression of the different components within the communities in the bow-tie structure in Fig. 5. Here, we selected communities whose community size equals at least $10$ firms. G-I indicates that both the GSCC and IN components were over-expressed in the communities. Similarly, G-O, G-T, I-O, I-T, and O-T represent over-expressions of GSCC-OUT, GSCC-TE, IN-OUT, IN-TE, and OUT-TE, respectively. This reflects the fact that most of the communities are solely composed of a particular component of the bow-tie structure. We also observe that there is a reasonable number of communities comprised of the combination of the GSCC and IN (i.e., the G-I component), which is also observed in the Japanese supply chain network~\\cite{chakraborty2018hierarchical}. This indicates that the flow of goods in the supply chain network is often confined within the GSCC and IN components as compared to any other combination of components of the bow-tie structure. Surprisingly, a large fraction of communities is in the TE component, not only supply but also procure any products and services from GSCC components. ”

#3-3. The nine claims about efficient supply chain networks are analyzed without addressing data on efficiency. 

-> In section “Introduction”, we added the following description. (line 62 in the revised manuscript)

“In E.~J.S.~Hearnshaw~{\\em et al.}'s study, the term ``efficient supply chain” was used to characterize the fact that a variety of resources such as financial, human, technological, and physical resources were effectively used in supply chains. ”

#3-4. The two tasks accomplished in the paper – demonstrating a bow-tie structure and addressing nine claims about efficient supply chain networks – are mixed together without clarifying the relationship between these two tasks. Should we think of a bow-tie structure within the framework of an efficient supply chain network? How do the nine claims relate to the bow tie structure?

-> Please see the response to Comment #1-1 for the related revisions.

#3-5. Many figures are un-related to the argument, and are only passingly mentioned in the main text (Fig. 5-7). The network figures are not styled consistently, and the edges are not clearly visible.

-> We elaborated the description of the overexpression networks as follows. (line 324 in the revised manuscript)

“To visualize the co-occurrences of node attributes such as firms location and primary industry within the communities, we construct a network of over-expressions as described below. We construct a weighted and undirected network of countries from their over-expression in communities with sizes larger than $100$ to show the interrelationships between countries. A link of weight one is placed between two countries if they over-express simultaneously within a community. Furthermore, we visualize the community structure of this network, as shown in Fig. 6. Here, communities are detected using the modularity maximization technique. This demonstrates that each community is formed by geographically closely located countries. 

The first community consists mainly of South America, the second of Eastern and Southeastern Asia and Oceania, the third of Western and Northern Europe, the fourth of Western Asia, the fifth of Eastern Europe, the sixth of North America, and the seventh of Northern Africa. This result reflects the trade relations among the countries. The countries that belong to the same community have significantly stronger ties of trade relations between them.

Similarly, we also constructed a weighted undirected network of over-expressed primary industries, where a link of weight w is present between two primary industries if they are over-expressed simultaneously in w communities. As can be seen in Fig. 7 below and Fig~S1 in Appendix S1, the clusters among the primary industries are formed based on their sector classifications. Here, communities are detected using the modularity maximization technique. The first community consists mainly of the industrial sector, the second of the consumer discretionary sector, the third of the health care sector, the fourth of the financials sector, the fifth of the energy and utility sectors, the sixth of the materials and industrials sector, the seventh of the information technology sector, and the eighth of the communication services. The result indicates the interactions of the flow of goods between the primary industries. Primary industries that belong to the same community depend heavily on each other. ”

-> We also improve the quality of Fig. 6 and Fig. 7 (in new numbering) to obtain better consistency and visibility.

#3-6. In general, this manuscript would need to clarify its goals much better in the introduction, and the empirical results would need to be interpreted more carefully. Results and various goals of the manuscript should be much better integrated into a coherent argument.

-> Please see the response to Comment #1-1 and #3-1 for the related revisions. In addition to the above, we revised section “conclusions” as follows. (line 447 in the revised manuscript)

“In this paper, we studied the topological properties of the global supply chain network to empirically verify the efficient supply chain propositions proposed by E.~J.S.~Hearnshaw~{\\em et al.}. We verified the propositions by conducting a large-scale empirical analysis of global supply chain data. The topological properties of the global supply chain network are empirically studied from three different angles: node-level characteristics, mesoscopic structural characteristics, and flow characteristics. An analysis from these three different angles was required to verify the efficient supply chain propositions and obtain a coarse-grained description of the global supply chain network in terms of its bow-tie and community structures.

The global supply chain data in the year 2017 were constructed by collecting various company data from Standard \\& Poor's Capital IQ platform. The total number of firms and directed links in our data were $437,453$ and $948,247$, respectively.

Degree distribution is characterized by a power law of the form with $\\gamma_{in} = 2.42$ and $\\gamma_{out} = 2.11$. The clustering coefficient decays $\\langle C(k) \\rangle \\sim k^{-\\beta_k}$ with an exponent $\\beta_k = 0.46$. This indicates the presence of a hierarchical structure to the supply chain network. We observed that $\\langle k_{nn}(k) \\rangle$ does not depend on $k$ and remains more or less constant with $k$, indicating the absence of nodal degree-degree correlation.

The bow-tie structure of the GWCC revealed that the OUT component was the largest and consisted of $41.1\\%$ of all firms. The GSCC component comprised $16.4\\%$ of all firms. We observed that upstream or downstream firms were located a few steps away from the GSCC. Then, we discovered the community structure of the network using the map equation method and characterized them according to their locations and industry classifications. We observed that the largest community was comprised of private firms mainly from the automotive retail sector based in the US. These firms are belong to the OUT component in the bow-tie structure of the global supply chain network. This indicates that the retail firms generally belong to the OUT component of the bow-ties structure.

Finally, we investigated the validity of the nine propositions on the supply chain network based on the results obtained from the topological properties. We confirmed the validity of Propositions S1 (i.e., short path length), Proposition S2 (i.e., power-law degree distribution), Proposition S3 (i.e., high clustering coefficient), Proposition S4 (i.e., ``fit-gets-richer” growth mechanism), Proposition S5 (i.e., truncation of power-law degree distribution), and Proposition S7 (i.e., community structure with overlapping boundaries) regarding the global supply chain network. However, the propositions related to link weight and the resilient nature of the network were not confirmed due to the limitations of our data and the narrow scope of the current study. Such analysis will be left for a future study. ”

---

## [Decision Letter · Decision Letter 1]

6 Jul 2020

PONE-D-20-05766R1

Verifying“efficient supply chain propositions”using topological characterization of the global supply chain network

PLOS ONE

Dear Dr. Ikeda,

Thank you for submitting your manuscript to PLOS ONE. After careful consideration, we feel that it has merit but does not fully meet PLOS ONE’s publication criteria as it currently stands. Therefore, we invite you to submit a revised version of the manuscript that addresses the points raised during the review process.

We look forward to receiving your revised manuscript.

Kind regards,

Hocine Cherifi

Academic Editor

PLOS ONE

Reviewers' comments:

Reviewer's Responses to Questions

**Comments to the Author**

1. If the authors have adequately addressed your comments raised in a previous round of review and you feel that this manuscript is now acceptable for publication, you may indicate that here to bypass the “Comments to the Author” section, enter your conflict of interest statement in the “Confidential to Editor” section, and submit your "Accept" recommendation.

Reviewer #1: (No Response)

Reviewer #2: (No Response)

Reviewer #3: All comments have been addressed

2. Is the manuscript technically sound, and do the data support the conclusions?

Reviewer #1: Yes

Reviewer #2: Yes

Reviewer #3: Yes

3. Has the statistical analysis been performed appropriately and rigorously? 

Reviewer #1: Yes

Reviewer #2: Yes

Reviewer #3: Yes

4. Have the authors made all data underlying the findings in their manuscript fully available?

Reviewer #1: Yes

Reviewer #2: Yes

Reviewer #3: No

5. Is the manuscript presented in an intelligible fashion and written in standard English?

Reviewer #1: Yes

Reviewer #2: Yes

Reviewer #3: Yes

6. Review Comments to the Author

Reviewer #1: The Authors have made a series of substantial changes to the text, the result of which is a significantly improved manuscript. The Authors have reflected on all of my previous concerns and I accept their answers and corrections in most cases. However, I have three minor points which still need to be addressed. I refer to them with the numbers in my previous report.

6. In line 111: how do these statistics prove the goodness of the data?

• I think I understand now, but the caption should also mention that total revenue on the vertical axis is measured for companies in the sample, not overall for the country.

11.-12. In line 216 the Authors draw attention to a difference between their results and previous ones. There should be a discussion of this difference. Similarly, in line 220 there is a difference revealed with respect to many other networks. There should be a discussion about the reasons of this difference.

• Still I feel that some discussion is missing, or at least clarifying the reasons why this discussion is not given here.

17. In line 319 the Authors say that clustering is moderate. But compared to what?

• This should be indicated in the text as well.

Reviewer #2: I think the authors have improved the quality of the manuscript by clarifying the aim of the paper.

However, I have few remarks:

- the network is directed as the authors affirm, but the authors decided to apply some topological indicators using the directed network, while others have been applied using an undirected network.

I think this choice must be justified. For instance, this is the case of community detection and clustering.

- the authors use the undirected version of clustering coefficient, but I think they must cite the clustering they have used (Watts and Strogatz or Wasserman and Faust). Additionally, although the authors decided to stay in the undirected case, they have to mention that in literature clustering coefficients for directed case exists and cite them (see

Fagiolo (2007), Clustering in Complex Directed Networks

Clemente, G.P. and Grassi, R. (2018) Directed clustering in weighted networks: a new perspective, Chaos, Solitons and Fractals.

Reviewer #3: Most points have been addressed by the authors, and the manuscript has a much clearer argumentation. However, one main concern that I had with the first version of the manuscript is still not sufficiently addressed. This main concern is the lack of conceptual integration of the two major ambitions of the manuscript (1: the bow-tie structure and 2: testing nine propositions of Hearnshaw et al.) The authors managed to integrate testing Hearnshaw et al. into the manuscript much better. But it is still not clear how the bow-tie finding integrates with this ambition. Does the bow-tie structure add a new topological requirement to having an efficient supply chain network? In that case, the bow-tie structure is a conceptual addition to Hearnshaw et al. Or is the bow-tie merely a consequence of Hearnshaw et al.’s requirements that they have not recognized? In that case the bow-tie structure is an empirical discovery that does not alter Hearnshaw et al.’s conceptualization.

These concerns are critical, as the own contribution of the manuscript is not clear. The ambition to test Hearnshaw et al. is clear now already in the abstract, but there is no mention about conclusions: what does this manuscript contribute? This is connected with the ambiguous language of “verifying”. While I recognize that there are multiple methodological traditions in science, I would believe that there is the broadest agreement that “verification” is never possible in the empirical sciences, and we should think about “testing” instead. A theory will never be verified, but rather tested, and found support for. I would strongly urge the authors to re-phrase “verification” for testing, with an indication of what aspects of the theory from Hearnshaw et al. they found support for. This could help clarify the contribution of the manuscript as well.

If I was the author, I would explore ways in which the novelty of the contribution can be identified, and highlighted already in the abstract.

As a more minor point, the network overexpression graphs for the country and industry level are much more readable, but their integration into the ambitions of the manuscript is still not clear. What aspect of Hearnshaw et al. do these talk to? I would not include almost two full page graphs that are part of only a sideline discussion, receiving only a passing mention in the text.

7. PLOS authors have the option to publish the peer review history of their article (what does this mean?). If published, this will include your full peer review and any attached files.

Reviewer #1: No

Reviewer #2: No

Reviewer #3: No

---

## [Author Response · Author response to Decision Letter 1]

6 Sep 2020

Reviewer #1: The Authors have made a series of substantial changes to the text, the result of which is a significantly improved manuscript. The Authors have reflected on all of my previous concerns and I accept their answers and corrections in most cases. However, I have three minor points which still need to be addressed. I refer to them with the numbers in my previous report.

Comment #1-1

6. In line 111: how do these statistics prove the goodness of the data?

• I think I understand now, but the caption should also mention that total revenue on the vertical axis is measured for companies in the sample, not overall for the country.

Response to the comment:

We added the sentence, “The vertical axis is aggregated for firms of each country in the global supply chain data.” to the caption of Fig. 1.

Comment #1-2

11.-12. In line 216 the Authors draw attention to a difference between their results and previous ones. There should be a discussion of this difference. Similarly, in line 220 there is a difference revealed with respect to many other networks. There should be a discussion about the reasons of this difference.

• Still I feel that some discussion is missing, or at least clarifying the reasons why this discussion is not given here.

Response to the comment:

line 278-283 in the revised paper

We added the following explanation for the small SCC in the global supply chain network.

In Japan, there is a large SCC in the ownership network [1]. This is well known as cross-shareholding, or “keiretsu” (a set of companies with interlocking business relationships and shareholdings) in Japanese. Correspondingly, there is a large SCC in the Japanese supply chain. On the other hand, the cross-shareholdings are not as pronounced in the world's firms as in Japan. As a result, the SCC is small in the global supply chain network.

[1] Haruka Kato, Hitomi Sato, Yuichi Kichikawa, Hiroshi Iyetomi, Wataru Souma, and Tsutomu Watanabe, Measurement of Value of Firms Based on Their Stock Ownership Relations, Book of Abstracts, the 8th Int'l Workshop on Complex Networks and their Applications (Lisbon, Portgal), 2019, pp. 450-452.

#1-3

17. In line 319 the Authors say that clustering is moderate. But compared to what?

• This should be indicated in the text as well.

Response to the comment:

Propositions S3: 

However, the value of the clustering coefficient is not too high (<<1). The observed moderate clustering coefficient indicates that Proposition S3: It has a high clustering coefficient. is weakly valid.

We changed these sentences in line 410-413 in the revised paper to as follows.

However, the value of the clustering coefficient is equivalent to many other networks, such as the power grid, mobile phone calls, science collaboration, etc. The observed moderate clustering coefficient indicates that Proposition S3: It has a high clustering coefficient. is valid.

 

Reviewer #2: I think the authors have improved the quality of the manuscript by clarifying the aim of the paper. However, I have few remarks:

Comment #2-1

- the network is directed as the authors affirm, but the authors decided to apply some topological indicators using the directed network, while others have been applied using an undirected network. I think this choice must be justified. For instance, this is the case of community detection and clustering.

Response to the comment:

We have used an undirected version of the network for measuring the clustering coefficient and average nearest neighbor degree to show the presence of hierarchical structure and degree mixing property of the network as the basic structural properties. 

For community structure, we have used a directed version of the network where the links represent flow of goods from suppliers to customers. 

At line 160 we added: We have used an undirected version of the network for measuring the clustering coefficient and average nearest neighbor degree to show the presence of hierarchical structure and degree mixing property of the network as the basic structural properties. 

We modified line 286-288: We employ the map equation method [13] to discover the communities in the GWCC of the global supply chain network using the directed nature of the links. The directed links represent flow of goods from suppliers to customers. 

Comment #2-2

- the authors use the undirected version of clustering coefficient, but I think they must cite the clustering they have used (Watts and Strogatz or Wasserman and Faust). Additionally, although the authors decided to stay in the undirected case, they have to mention that in literature clustering coefficients for directed case exists and cite them (see Fagiolo (2007), Clustering in Complex Directed Networks, Clemente, G.P. and Grassi, R. (2018) Directed clustering in weighted networks: a new perspective, Chaos, Solitons and Fractals.)

Response to the comment:

Line 151 in the revised paper: To calculate the clustering coefficient C(k), we considered a simple undirected version of the network. 

In the manuscript, we have added the following lines for the clustering coefficient at line 151-157: 

The clustering coefficient C_i of i th node is defined as [1], 

C_i = 2 E_i / k_i(k_i - 1), 

where E_i is the number of links between all the k_i neighbors of i. We have measured the clustering coefficient <C (k)> averaged over the subset of nodes of degree k. Further one can measure more complex clustering coefficient considering the directed nature of links [2]. However, we only calculate the simple undirected clustering coefficient to show the basic structural property of the network. 

[1] D. J. Watts and Steven Strogatz (June 1998). "Collective dynamics of 'small-world' networks". Nature. 393 (6684): 440–442

[2] Fagiolo (2007), Clustering in Complex Directed Networks, Clemente, G.P. and Grassi, R. (2018) Directed clustering in weighted networks: a new perspective, Chaos, Solitons and Fractals.

 

Reviewer #3: Most points have been addressed by the authors, and the manuscript has a much clearer argumentation. However, one main concern that I had with the first version of the manuscript is still not sufficiently addressed. 

Comment #3-1

This main concern is the lack of conceptual integration of the two major ambitions of the manuscript (1: the bow-tie structure and 2: testing nine propositions of Hearnshaw et al.) The authors managed to integrate testing Hearnshaw et al. into the manuscript much better. But it is still not clear how the bow-tie finding integrates with this ambition. Does the bow-tie structure add a new topological requirement to having an efficient supply chain network? In that case, the bow-tie structure is a conceptual addition to Hearnshaw et al. Or is the bow-tie merely a consequence of Hearnshaw et al.’s requirements that they have not recognized? In that case the bow-tie structure is an empirical discovery that does not alter Hearnshaw et al.’s conceptualization.

These concerns are critical, as the own contribution of the manuscript is not clear. The ambition to test Hearnshaw et al. is clear now already in the abstract, but there is no mention about conclusions: what does this manuscript contribute? This is connected with the ambiguous language of “verifying”. While I recognize that there are multiple methodological traditions in science, I would believe that there is the broadest agreement that “verification” is never possible in the empirical sciences, and we should think about “testing” instead. A theory will never be verified, but rather tested, and found support for. I would strongly urge the authors to re-phrase “verification” for testing, with an indication of what aspects of the theory from Hearnshaw et al. they found support for. This could help clarify the contribution of the manuscript as well. If I was the author, I would explore ways in which the novelty of the contribution can be identified, and highlighted already in the abstract.

Response to the comment:

I appreciate the excellent suggestion. I agree to re-phrase “verification” for “testing.” However, to my knowledge, in the natural sciences, the word “verify” is sometimes used in the same sense as “test.” For example, “experimental verification” is found in many literature.

We added the following sentences after Line 383 in the revised paper and changed the abstract and conclusions.

Original propositions of Hearnshaw S1 just mentioned a short characteristics path length. As described in the discussion of Table 4, the upstream or downstream firms are located only a few steps away from the GSCC. The short path from the GSCC to IN and OUT was only discovered by analyzing the bow-tie structure. Therefore, the short path length in the bow-tie structure is a conceptual addition to the original propositions of Hearnshaw.

Comment #3-2

As a more minor point, the network overexpression graphs for the country and industry level are much more readable, but their integration into the ambitions of the manuscript is still not clear. What aspect of Hearnshaw et al. do these talk to? I would not include almost two full page graphs that are part of only a sideline discussion, receiving only a passing mention in the text.

Response to the comment:

We added the following sentences in Line 455-463 in the revised paper. 

The over-expression of countries and sectors in the large communities shown in Table 5 allows us to characterize community formation by two factors. One is the over-expression of countries in the large communities shown in Fig. 6. The other is the over-expression network of sectors shown in Figure 7. In these figures, a country tends to form communities in neighboring countries, and a sector tends to form communities in the same industry. This result suggests that community formation is due to multiple factors, and the over-expression networks in Fig.6 and Fig.7 provide support for the appropriateness of the overlapping boundary of community formation in Proposition S7.

---

## [Editor Report · Decision Letter 2]

11 Sep 2020

Testing“efficient supply chain propositions”using topological characterization of the global supply chain network

PONE-D-20-05766R2

Dear Dr. Ikeda,

We’re pleased to inform you that your manuscript has been judged scientifically suitable for publication and will be formally accepted for publication once it meets all outstanding technical requirements.

Kind regards,

Hocine Cherifi

Academic Editor

PLOS ONE
---

## [Editor Report · Acceptance letter]

22 Sep 2020

PONE-D-20-05766R2 

Testing “efficient supply chain propositions” using topological characterization of the global supply chain network

Dear Dr. Ikeda:

I'm pleased to inform you that your manuscript has been deemed suitable for publication in PLOS ONE. Congratulations! Your manuscript is now with our production department. 

Kind regards, 

on behalf of

Professor Hocine Cherifi 

Academic Editor

PLOS ONE